# Epidemiological investigation and surgical treatment of canine mammary tumors in Dalian, China, from 2019 to 2023

Zheng Jing[1]*, Jiawang Feng[2], Hongyan Jin[2]*

1 Jiangsu Agri-animal Husbandry Vocational College, Taizhou, Jiangsu, China, 2 Tibet Vocational Technical College, Lasa, Tibet, China

* jing99sam@163.com (ZJ); jhy780401@126.com (HJ)

**Data Availability Statement:** All relevant data are within the manuscript and its Supporting Information files.

## Abstract

Objective of this study is to investigate the epidemiological characteristics, clinical features, and treatment outcomes of canine mammary tumors in Dalian, providing insights into prevention and management strategies. A retrospective analysis was conducted on 198 cases of canine mammary tumors diagnosed in outpatient departments across several veterinary hospitals in Dalian. Data on breed, age, sex, tumor location, and clinical staging were collected and correlated with treatment modalities and prognosis. Poodles, Chinese pastoral dogs, and Cocker Spaniels exhibited higher incidence rates. The majority of affected dogs were middle-aged and older females, with unneutered dogs and those with a history of false pregnancies being at the highest risk. Benign tumors were more common in younger dogs, while malignant tumors predominated in older dogs, accounting for 89.9% of the cases. Early surgical intervention significantly improved survival and quality of life. Early detection, prompt surgical treatment, and post-operative follow-up are essential for optimal outcomes in canine mammary tumor management. This study summarizes the impact of early sterilization on tumor development and suggests that preventive measures, such as total ovarian extraction prior to the first estrus, are effective in reducing the incidence of mammary tumors.

## Introduction

Mammary tumors are a prevalent health issue in canines, with canine mammary tumors (CMTs) representing a significant proportion of neoplastic diseases in female dogs [1]. Epidemiological surveys indicate that CMTs account for 25% to 42% of tumors in susceptible female dogs, with over 50% of these being malignant [2]. The incidence of mammary tumor in dogs is significantly higher than that in other domestic animals, with a notable increase observed from juvenile to geriatric stages, peaking in females aged 7 to 13 years with an average age of 8.6 years, while it is rarely seen in male dogs [3]. Recent studies have further elucidated the prevalence of CMTs, with 13.4% of all canine tumors being mammary in origin, predominantly affecting middle-aged, unneutered females, particularly between the ages of 6 and 10 [4]. Male

**Funding:** This study received funding from School level scientific research project of Jiangsu Agri-animal Husbandry Vocational College (grant number: NSF2023CB14). The funders had no role in study design, data collection and analysis, decision to publish, or preparation of the manuscript.

**Competing interests:** The authors have declared that no competing interests exist.

dogs are rare, with an incidence of $<1\%$, and the incidence of female dogs is 62 times that of male dogs with a recent study revealing that the majority of male CMTs are benign and amenable to long-term remission following surgical intervention [4]. Male dogs may develop supportive cell tumors that secrete estrogen. Early ovidectomization can effectively prevent the incidence of breast tumors. Domestic investigations show that the most frequent breeds of mammary tumors in dogs are mainly small lion dogs, German Shepherd dogs, horse dogs, etc [5].

The diagnosis of canine breast tumors is usually made by the following methods: (1) history investigation, physical examination, laboratory examination, cytology examination, etc. (2) imaging examination, including X-ray examination, ultrasonic scanning examination, CT examination and nuclear magnetic resonance technology. (3) Subsequently histopathological diagnosis is conducted to confirm the nature of the tumor. In the process of tumor detection, various examination methods have their certain role, among which histopathological diagnosis is still one of the commonly used and accurate methods in the clinical diagnosis and treatment of breast tumors [5,6]. In the therapeutic landscape, a variety of approaches have been employed, including surgery, radiation therapy, chemotherapy, immunotherapy, hormone therapy, hyperthermia, cryotherapy, and traditional Chinese medicine [7]. Surgical therapy stands as the primary modality for CMT management, complemented by adjuvant therapies such as hormone and chemotherapy, which synergize to enhance treatment efficacy.

This study presents a comprehensive analysis of 198 CMT cases from animal hospitals in Dalian between 2019 and 2023, thereby informing about examining the interplay between breed, age, sex, neuter status, diet, and tumor location. Our aim is to explore the epidemiological patterns and risk factors associated with CMTs, thereby informing more targeted diagnostic, therapeutic, and preventive strategies. Additionally, by evaluating the postoperative outcomes of surgical interventions, this study seeks to contribute valuable insights to the clinical management of canine mammary tumors.

## Materials and methods

### Clinical case collection

From 2019 to 2023, a retrospective study was conducted on 198 cases of canine mammary tumors (CMTs) from domestic dogs presented at a veterinary hospital in Dalian, northeast China. Ethical approval was not required as the study involved analysis of existing clinical data. The collected data encompassed age, breed, gender, body conformation, spayed status, histopathological diagnosis, tumor count, size, and location. Histopathological examination was performed following standard procedures including formalin fixation, paraffin embedding, hematoxylin-eosin (HE) staining, and evaluation under optical microscopy.

### Epidemiological analysis of canine mammary tumors

To explore correlations between epidemiological factors (age, breed, gender, and spaying status) and tumor characteristics (size, number, and location) in conjunction with histological diagnoses. Age was stratified into four groups: 0–4 years, 5–8 years, 9–12 years, and over 13 years. Tumor size was categorized according to the World Health Organization (WHO) classification: T1 ($<3$ cm), T2 (3–5 cm), and T3 ($>5$ cm) [8]. Breeds were categorized as purebred or hybrid, with hybrid size further classified into small, medium, and large based on height, aligning with the International Canine Federation (FCI) standards. Dogs were also grouped based on the number of tumors and the specific location within the mammary gland, categorized into five pairs. Tumor malignancy was classified following the system, developed by

Misdorp et al, distinguishing between benign and malignant tumors with malignancy levels graded as I, II, and III [9,10].

## Surgical procedures

The operation was performed by Dr. Zheng Jing, who obtained the Chinese Professional Veterinary Doctor qualification Certificate (Certificate number: A012015210149) in 2015. He has been engaged in surgical operations such as tumor resection for 12 years and has rich clinical experience. Prior to surgery, a thorough understanding of the anatomical structure of the mammary glands, including blood and lymphatic distribution, was established. Surgical plans were tailored based on tumor size and location, employing various resection methods such as lumpectomy, mastectomy, local mastectomy, unilateral mastectomy, and bilateral radical mastectomy. The goal was to minimize incisions, remove as much tumor tissue and surrounding tissue as possible to prevent recurrence and metastasis, and to optimize patient recovery. Postoperatively, dogs were fitted with Elizabethan collars to prevent wound licking and biting, and postoperative care was intensified, including dietary management. The specific surgical procedures are as follows: First, preoperative examination, anesthesia, and operation in strict accordance with surgical operation specifications. Subsequently, the surgical area was disinfected, the skin was cut, the subcutaneous tissue was removed, and the uterus and ovary were removed first. Then the breast and the skin covering the breast area are removed together according to the above principles of tumor removal. During the operation, the heart rate, blood pressure and respiration were closely monitored. Finally, hemostasis is sufficient to suture the skin and subcutaneous tissue.

## Statistical analyses

The analysis of data was performed using descriptive statistics, as well as bivariate and multivariate analyses. Data were analyzed using Prism 7.0 (Graph Pad Inc., USA), and the value of $p < 0.05$ was considered significant.

## Results

### Breed-specific prevalence of canine mammary tumors

The 198 cases of canine mammary tumors (CMTs) were distributed across 19 breeds, with Poodles, Chinese rural dogs, and Cocker Spaniels showing the highest prevalence rates of 35.4%, 17.7%, and 10.1%, respectively. Notably, Pomeranians, Beagles, and Chow Chows also exhibited relatively high prevalence rates of 6.1%, 6.1%, and 5.1%, respectively. These findings suggest a breed-specific susceptibility to CMTs (Table 1).

### Age distribution in canine mammary tumor onset

The age of affected dogs ranged from 1 to 20 years, with the majority of cases (49.0%) occurring in the 5–8 years age group. Notably, the age group of 8 years old had the highest number of cases (36 cases). Dogs over 8 years old accounted for 43.4% of the total cases, with the 9-year age group being the most affected within this category. The incidence of CMTs in dogs around the age of 8 was found to be the highest, indicating a correlation between age and tumor development (Table 2).

### Classification and characteristics of canine mammary tumors

Laboratory analysis revealed that of the 198 CMT cases, 20 were benign, representing 10.1% of the total. A higher prevalence of benign tumors was observed in younger dogs, exemplified by

**Table 1. Statistical results of breast tumor cases in different breeds of dogs.**

| breeds | Case (Number) | Percentage (%) |
|---|---|---|
| Poodle | 70 | 35.4 |
| Chinese Pastoral Dog | 35 | 17.7 |
| Cocker Dog | 20 | 10.1 |
| pomeranian | 12 | 6.1 |
| Bichon | 12 | 6.1 |
| Chow Lion Dog | 10 | 5.1 |
| Wolf Green Dog | 6 | 3 |
| Pekingese | 7 | 3.5 |
| Chihuahua Dog | 5 | 2.5 |
| Little Deer Dog | 5 | 2.5 |
| Samoyed dog | 4 | 2 |
| Silver Fox Dog | 2 | 1 |
| Yorkshire Terrier | 1 | 0.5 |
| Schnauzer | 1 | 0.5 |
| German Shepherd | 2 | 1 |
| Golden Retriever | 2 | 1 |
| Maltese dog | 2 | 1 |
| Shetland Sheepdog | 1 | 0.5 |
| Labrador dog | 1 | 0.5 |

a benign ductal papilloma in a 1-year-old Cocker Spaniel. Malignant tumors predominated, with 178 cases (89.9%), and their incidence increased with age. The oldest case involved a 20-year-old Chinese garden dog with a grade III malignant adenocarcinoma that had

**Table 2. Incidence of mammary tumors in dogs of different ages.**

| ages | Case (Number) | Percentage (%) |
|---|---|---|
| 1 | 1 | 0.5 |
| 2 | 2 | 1 |
| 3 | 8 | 4 |
| 4 | 5 | 2.5 |
| 5 | 17 | 8.6 |
| 6 | 19 | 9.6 |
| 7 | 25 | 12.6 |
| 8 | 36 | 18.2 |
| 9 | 27 | 13.6 |
| 10 | 13 | 6.6 |
| 11 | 10 | 5.1 |
| 12 | 14 | 7.1 |
| 13 | 11 | 5.6 |
| 14 | 2 | 1 |
| 15 | 4 | 2 |
| 16 | 1 | 0.5 |
| 17 | 2 | 1 |
| 18 | 0 | 0 |
| 19 | 0 | 0 |
| 20 | 1 | 0.5 |

**Table 3. Statistics of frequently occurring sites of canine breast tumors.**

| Type | Total (number) | classification | Case (number) | Percentage(%) |
|---|---|---|---|---|
| benign | 20 | Ductal papilloma | 18 | 9.1 |
| | | adenoma | 2 | 1 |
| Malignant | 178 | Compound carcinoma of breast | 100 | 50.5 |
| | | Breast adenocarcinoma (grade I) | 66 | 33.3 |
| | | Breast adenocarcinoma (grade II) | 9 | 4.6 |
| | | Breast adenocarcinoma (grade III) | 3 | 1.5 |

metastasized, severely impacting the dog's quality of life and necessitating surgical intervention (Table 3).

## Surgical treatment outcomes

In the 198 CMT surgeries, 185 cases underwent unilateral tumor resection, with 99 cases on the right side and 86 on the left. Benign tumors were confirmed in 20 cases via histopathological diagnosis. Among these, 15 cases underwent timely ovarian hysterectomy, with no recurrence observed postoperatively. For malignant tumors, 165 cases were recorded, with 92 being compound tumors. Of these, 83 showed no obvious metastasis, while 11 cases exhibited varying degrees of metastasis within 4–10 months post-surgery. Post-surgical outcomes varied, with some cases showing long-term survival and others succumbing to metastasis or non-tumor causes. Histopathological grading post-operation revealed 64 cases of grade I tumors, with 4 cases experiencing recurrence within six months. These cases underwent further surgery with good postoperative recovery and no observed metastasis to date. The remaining cases showed no evidence of tumor recurrence beyond 12 months post-surgery (Fig 1).

## Microscopic features

As depicted in Fig 2A, the tumor exhibits indistinct margins and is marked by a substantial proliferation of epithelial cells. A subset of these cells are luminal epithelial cells, which the neoplastic cells are arranged into irregularly shaped and sized glandular structures, ranging from a single layer to multiple layers. The luminal cells can also form papillary projections. The atypical epithelioid tumor cells demonstrate high-grade atypia, with eosinophilic or vacuolated cytoplasm, and nuclei that are ovoid to round in shape. The chromatin is described as punctate to coarse, with the presence of 1–2 nucleoli. Mitotic figures within the tumor cells range from 0 to 2, and the interior reveals multifocal necrosis along with numerous tumor thrombi. Consequently, the diagnosis is a moderately malignant breast adenocarcinoma (Grade II), which has likely metastasized. The prognosis is considered guarded to poor.

As shown in Fig 2B, the tumor cells exhibit a diffuse infiltrative pattern with moderate cellular and nuclear atypia. There are relatively few mitotic figures, and the pleomorphism of the cells and nuclei ranges from moderate to pronounced. The tumor cells are also infiltrated by a large number of eosinophilic granulocytes, and there are areas of necrosis within the tumor. It is diagnosed as a granular cell tumor, Grade II-III, with a guarded to poor prognosis.

In Fig 2C, the tumor exhibits dilated glandular structures filled with red fluid. The stroma shows fibrous tissue proliferation. The tumor cells are predominantly spindle-shaped, forming cellular clusters with a fascicular arrangement. Numerous mitotic figures are observed. It is diagnosed as a grade III phyllodes tumor of the breast, with a poor prognosis. It is recommended to perform adjuvant chemotherapy after excision.

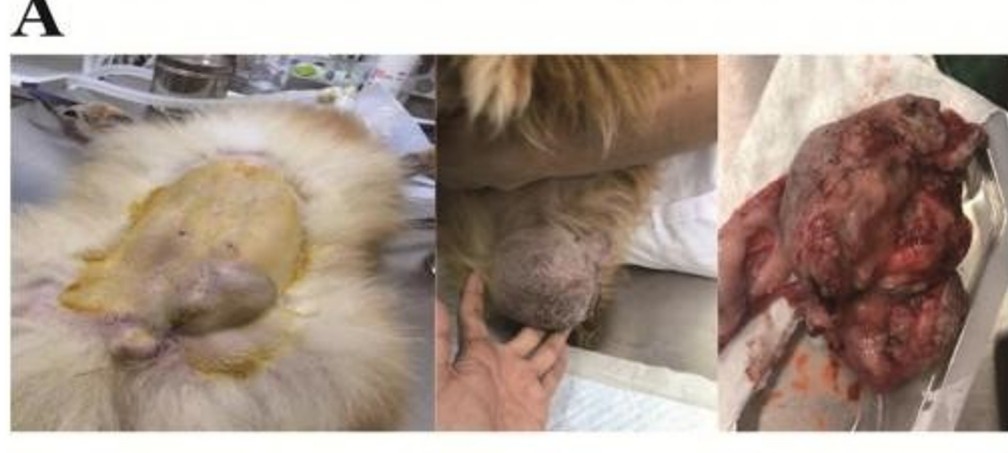

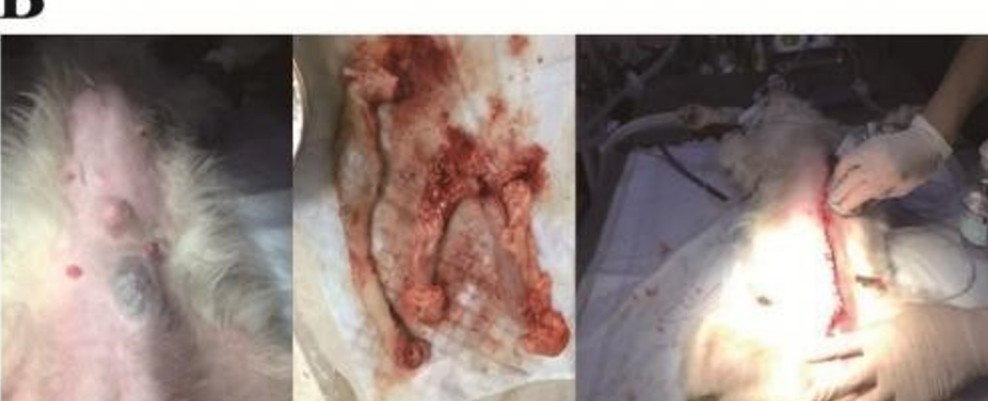

**Fig 1. Typical cases of canine breast tumor.** (A) 8-year-old Pomeranian dog 'Yuanyuan' had a mammary tumor in the inguinal mammary area, which was surgically removed. (B) 6-year-old Samoyed dog 'Gouzi' underwent bilateral mammary tumor removal and spaying.

In Fig 2D, the tumor lacks a fibrous capsule and has clear boundaries, representing a multi-focal malignant breast tumor. The tumor cells are characterized by the proliferation of both ductal and myoepithelial elements, with metaplastic changes in the epithelial cells also leading to the presence of focal osseous tissue distributed within the breast tissue. The epithelial cells are arranged in acinar tubules and cell cords, with some showing distinct breast duct lumens and a mixture of secretions and fibrin. The cells are cuboidal or cylindrical, with a moderate amount of eosinophilic material in the cytoplasm. The nuclei are round or oval, with finely stained chromatin and one or more nucleoli centrally located. There is moderate heterogeneity in cell and nuclear size, and mitotic figures are present. Therefore, it is diagnosed as a grade II mixed epithelial carcinoma of the breast. It is recommended to perform a total mastectomy (or segmental resection), along with a peripheral lymph node dissection and a comprehensive clinical staging of the tumor.

### Postoperative cure and recurrence rates

Analysis of postoperative outcomes revealed a 100% cure rate following benign tumor resection. For malignant tumors, unilateral resection achieved a 100% cure rate. The recurrence

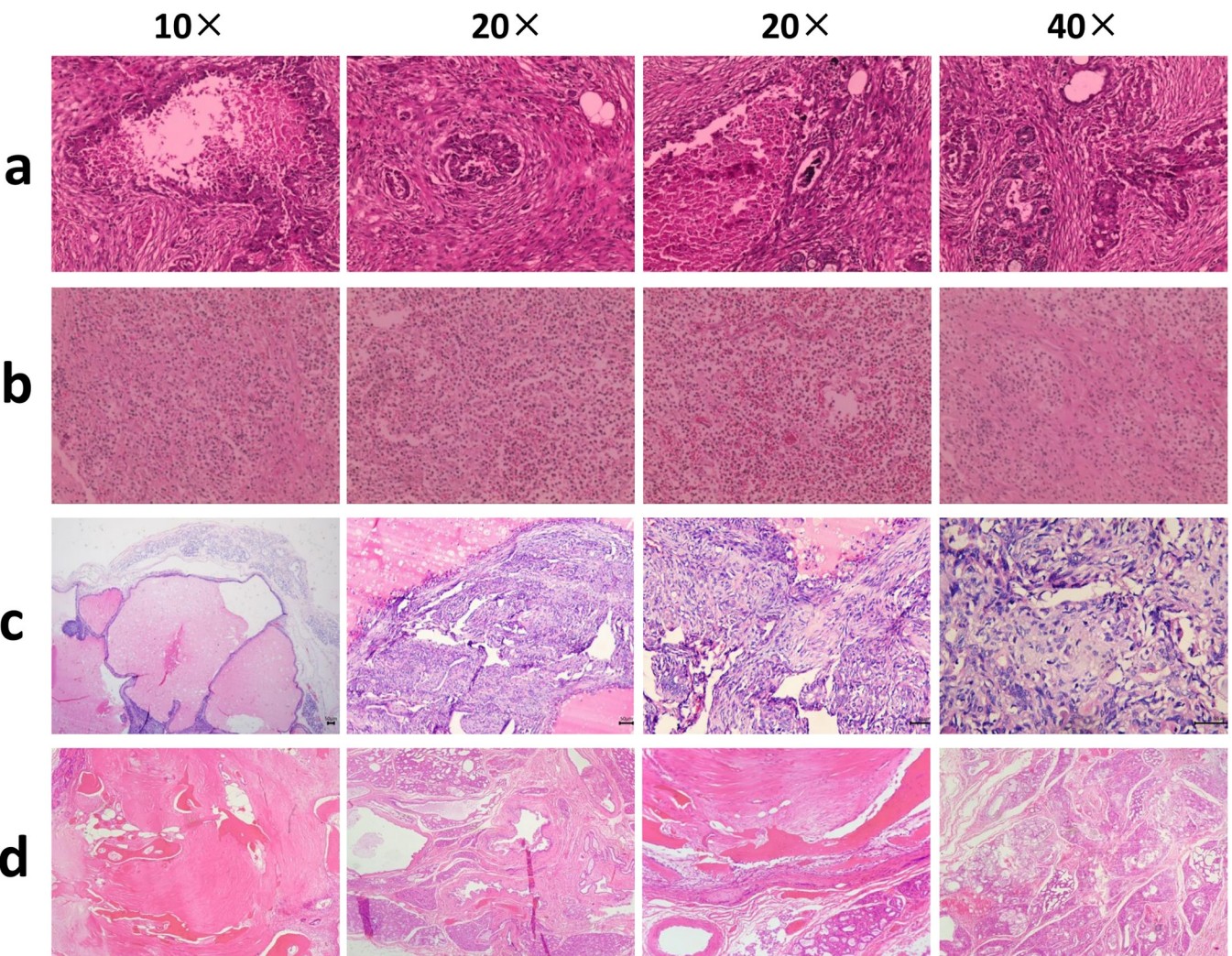

**Fig 2. Histopathological features of canine mammary tumors.** (a) Mammary gland adenocarcinoma. (b) Mast cell tumor. (c) multiple mammary adenomas. (d) Fibrosarcoma of breast. (e) Complex adenoma of breast. (f) Mammary mixed type carcinoma.

rate in cases of local tumor resection was associated with age and timing of sterilization, with a 100% cure rate observed in dogs sterilized under 5 years of age. In contrast, the recurrence rate for dogs sterilized and undergoing malignant tumor resection over 5 years old was 53.5%. The recurrence rate for patients undergoing total mastectomy and older than 5 years was 82.1%.

## Discussion

Canine mammary tumors (CMTs) constitute a diverse array of aberrant cellular proliferations, distinguished by their rapid and disorganized growth dynamics, which underscore the genetic complexity inherent to these conditions [3,11–13]. The pathogenesis of CMTs is intricate and multifactorial, encompassing a spectrum of environmental carcinogens, physiological determinants, and genetic predispositions of the host. Environmental contributors to CMT development include exposure to a myriad of chemical, physical, and biological carcinogens, alongside the impact of climate variability [14]. Concurrently, a suite of intrinsic factors such as species characteristics, breed-specific traits, age, sex, dietary practices, the timing of

sterilization procedures, hormonal milieu, and immune status are implicated in the tumorigenesis process [15]. The evolution of CMTs is best described as a protracted trajectory, punctuated by the progressive accumulation of genetic alterations [6]. These mutations arise from the complex interplay between an individual's inherent biology and external environmental pressures, culminating in the phenotypic drift from normal to neoplastic cellular states.

Our study affirms the breed-specific susceptibility to canine mammary tumors (CMTs), echoing prior research that identified certain breeds, such as Poodles and Cocker Spaniels, as having elevated incidence rates. The genetic architecture of different breeds is likely to play a pivotal role in their varying propensities for developing CMTs, with a descending order of vulnerability observed for breeds including Boxers, Shorthaired Hounds, and Rottweilers, etc. In our survey of 198 canine cases, Poodles, Chinese garden dogs, and Cocker Spaniels were notably among those with the highest prevalence rates, succeeded closely by Pomeranians, Bichon Frises, and Griffons. These observations, while distinct from some international findings, are consistent with a general trend indicating a heightened risk of CMTs in smaller breeds. The genetic and environmental factors that might account for these breed-specific differences warrant further exploration to elucidate the underlying mechanisms contributing to CMT development.

Age is also an important factor affecting the prevalence of breast cancer in dogs. A survey of dogs suffering from breast tumors showed that the incidence rate was 1% at the age of 6, 6% at the age of 8, and 13% at the age of 10 [16]. Our findings, which include the youngest case at 1 year and the oldest at 20 years, indicate a pronounced peak in the 5–8 years age bracket, where 97 cases accounted for 49% of our total cohort. Notably, the age group of 8 years old was most significantly affected, with 36 cases representing a substantial subset of this age-related peak. While 85 cases, constituting 43.4%, were dogs over 8 years old, with the age of 9 years old showing the highest incidence within this group, the representation of dogs under 5 years was considerably lower, with 16 cases at 8.1%, including 8 cases at the age of 3 years. This age-related distribution underscores the heightened vulnerability of middle-aged to older female dogs to CMTs. The observed age-related incidence aligns with previous studies, which report an incremental risk with advancing age [9,17]. However, the presence of CMTs in younger dogs suggests that contemporary environmental and societal factors, such as the pervasive use of chemical and hormonal substances, may be influencing endocrine balance and immune function in dogs. These factors could potentially disrupt normal physiological processes, leading to a dysregulation of hormone secretion and an increased susceptibility to neoplastic diseases.

The treatment of canine breast tumor is preferably started in the early and early stage. Once the breast tumor metastasis occurs, even if the breast tumor is removed by surgery, the quality of life of the affected dog can only be temporarily improved, and it cannot be cured. Early breast tumor resection combined with hysterectomy and ovariectomy plays a positive role in improving the cure rate, prolonging the postoperative survival time and improving the quality of life of dogs [18]. In terms of the prevention of canine breast tumor, total ovariectomy and hysterectomy can be performed before the first estrus of the dog, which will greatly reduce the incidence of breast tumor, improve the quality of life of the dog and prolong its life [19]. Therefore, prevention is the most important. Once breast tumor occurs, it should be recognized, diagnosed and treated early.

## Conclusions

This study found that the prevalence of breast cancer in poodle, Chinese shepherd dog and Cocker Spaniel was higher than that in other dog breeds. The older the dog, the higher the

probability of breast cancer. Early detection and diagnosis, early completion of breast tumor resection, and regular postoperative follow-up will obtain the best therapeutic effect of breast tumor, while the prevention of breast tumor is still the main. Total ovariectomy before estrus in dogs has the best preventive effect. This study provides new ideas and strategies for the prevention and treatment of canine breast tumors in Northeast China.

## Supporting information

**S1 Table. Case information collected in this study.**
(XLSX)

## Acknowledgments

We thank International Science Editing for language editing during the preparation of this manuscript.

## Author Contributions

**Conceptualization:** Zheng Jing, Hongyan Jin.

**Data curation:** Zheng Jing.

**Formal analysis:** Zheng Jing.

**Funding acquisition:** Zheng Jing.

**Methodology:** Jiawang Feng.

**Project administration:** Hongyan Jin.

**Resources:** Jiawang Feng.

**Software:** Jiawang Feng.

**Supervision:** Hongyan Jin.

**Writing – original draft:** Zheng Jing.

**Writing – review & editing:** Hongyan Jin.

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
