## [Decision Letter · Decision Letter 0]

25 Oct 2024

PONE-D-24-31787Epidemiological Investigation and Surgical Treatment of Canine Mammary Tumors in Dalian, China, from 2019 to 2023PLOS ONE

Dear Dr. Jing,

Thank you for submitting your manuscript to PLOS ONE. After careful consideration, we feel that it has merit but does not fully meet PLOS ONE’s publication criteria as it currently stands. Therefore, we invite you to submit a revised version of the manuscript that addresses the points raised during the review process.

We look forward to receiving your revised manuscript.

Kind regards,

Sudheesh Sreerangam Nair, B.V.Sc &A.H., M.V.Sc., Ph.D

Academic Editor

PLOS ONE

Journal requirements: When submitting your revision, we need you to address these additional requirements. 1. Please ensure that your manuscript meets PLOS ONE's style requirements, including those for file naming. The PLOS ONE style templates can be found at https://journals.plos.org/plosone/s/file?id=wjVg/PLOSOne_formatting_sample_main_body.pdf and https://journals.plos.org/plosone/s/file?id=ba62/PLOSOne_formatting_sample_title_authors_affiliations.pdf 2. We suggest you thoroughly copyedit your manuscript for language usage, spelling, and grammar. If you do not know anyone who can help you do this, you may wish to consider employing a professional scientific editing service.  The American Journal Experts (AJE) (https://www.aje.com/) is one such service that has extensive experience helping authors meet PLOS guidelines and can provide language editing, translation, manuscript formatting, and figure formatting to ensure your manuscript meets our submission guidelines. Please note that having the manuscript copyedited by AJE or any other editing services does not guarantee selection for peer review or acceptance for publication.  Upon resubmission, please provide the following: The name of the colleague or the details of the professional service that edited your manuscript A copy of your manuscript showing your changes by either highlighting them or using track changes (uploaded as a *supporting information* file) A clean copy of the edited manuscript (uploaded as the new *manuscript* file)”. 3. PLOS requires an ORCID iD for the corresponding author in Editorial Manager on papers submitted after December 6th, 2016. Please ensure that you have an ORCID iD and that it is validated in Editorial Manager. To do this, go to ‘Update my Information’ (in the upper left-hand corner of the main menu), and click on the Fetch/Validate link next to the ORCID field. This will take you to the ORCID site and allow you to create a new iD or authenticate a pre-existing iD in Editorial Manager. 4. Thank you for stating the following financial disclosure:  [School level scientific research project of Jiangsu Agri-animal Husbandry Vocational College (grant number: NSF2023CB14) . ].  Please state what role the funders took in the study.  If the funders had no role, please state: ""The funders had no role in study design, data collection and analysis, decision to publish, or preparation of the manuscript."" If this statement is not correct you must amend it as needed. Please include this amended Role of Funder statement in your cover letter; we will change the online submission form on your behalf. 5. We note that you have indicated that there are restrictions to data sharing for this study. PLOS only allows data to be available upon request if there are legal or ethical restrictions on sharing data publicly. For more information on unacceptable data access restrictions, please see http://journals.plos.org/plosone/s/data-availability#loc-unacceptable-data-access-restrictions.  Before we proceed with your manuscript, please address the following prompts: a) If there are ethical or legal restrictions on sharing a de-identified data set, please explain them in detail (e.g., data contain potentially identifying or sensitive patient information, data are owned by a third-party organization, etc.) and who has imposed them (e.g., a Research Ethics Committee or Institutional Review Board, etc.). Please also provide contact information for a data access committee, ethics committee, or other institutional body to which data requests may be sent. b) If there are no restrictions, please upload the minimal anonymized data set necessary to replicate your study findings to a stable, public repository and provide us with the relevant URLs, DOIs, or accession numbers. For a list of recommended repositories, please seehttps://journals.plos.org/plosone/s/recommended-repositories. You also have the option of uploading the data as Supporting Information files, but we would recommend depositing data directly to a data repository if possible. We will update your Data Availability statement on your behalf to reflect the information you provide. 6. In the online submission form, you indicated that [The datasets presented in this article are not readily available, and requests to access the datasets should be directed to ZJ, jing99sam@163.com.]. All PLOS journals now require all data underlying the findings described in their manuscript to be freely available to other researchers, either 1. In a public repository, 2. Within the manuscript itself, or 3. Uploaded as supplementary information.This policy applies to all data except where public deposition would breach compliance with the protocol approved by your research ethics board. If your data cannot be made publicly available for ethical or legal reasons (e.g., public availability would compromise patient privacy), please explain your reasons on resubmission and your exemption request will be escalated for approval.  7. Please remove your figures from within your manuscript file, leaving only the individual TIFF/EPS image files, uploaded separately. These will be automatically included in the reviewers’ PDF.

Reviewers' comments:

Reviewer's Responses to Questions

**Comments to the Author**

1. Is the manuscript technically sound, and do the data support the conclusions?

Reviewer #1: Yes

Reviewer #2: No

2. Has the statistical analysis been performed appropriately and rigorously? 

Reviewer #1: Yes

Reviewer #2: I Don't Know

3. Have the authors made all data underlying the findings in their manuscript fully available?

Reviewer #1: Yes

Reviewer #2: No

4. Is the manuscript presented in an intelligible fashion and written in standard English?

Reviewer #1: Yes

Reviewer #2: Yes

5. Review Comments to the Author

Reviewer #1: Great study and interesting findings! I have a few suggestions for building some sentences differently:

Abstract: Objective of this study is to investigate...

Line 63: Subsequently histopathological diagnosis is conducted to confirm the nature of the tumor.

L 73: thereby informing about...

L 91: Our aim is to explore...

L 100: Developed by Misdorp et al.

L 185: Analysis of postoperative outcomes revealed...

L 237: The treatment of canine breast tumor is prefereably started in the early stage.

L 246: Should be recognized, diagnosed and treated early.

Reviewer #2: I would say that the statistics unfounded

Surgical procedure was not explicitly described

Qualifications and expertise of the surgeons were not mentioned

Pictures of surgical procedures are not shown to see glitches in the procedure

6. PLOS authors have the option to publish the peer review history of their article (what does this mean?). If published, this will include your full peer review and any attached files.

Reviewer #1: **Yes: **Silvia Gardeweg

Reviewer #2: No

---

## [Author Response · Author response to Decision Letter 0]

6 Nov 2024

Responses to reviewers letter

Responses to the editor's comments:

Q1. Please ensure that your manuscript meets PLOS ONE's style requirements, including those for file naming. The PLOS ONE style templates can be found at 

 A1: The manuscript has been revised according to the plos one journal template above.

Q2. We suggest you thoroughly copyedit your manuscript for language usage, spelling, and grammar. If you do not know anyone who can help you do this, you may wish to consider employing a professional scientific editing service. 

 A clean copy of the edited manuscript (uploaded as the new *manuscript* file)”.

 A2: International Science Editing provided language editing in the preparation of the revised manuscript. The “Revised Manuscript with Track Changes” and “clean manuscript” files have been prepared as required.

Q3. PLOS requires an ORCID iD for the corresponding author in Editorial Manager on papers submitted after December 6th, 2016. Please ensure that you have an ORCID iD and that it is validated in Editorial Manager. To do this, go to ‘Update my Information’ (in the upper left-hand corner of the main menu), and click on the Fetch/Validate link next to the ORCID field. This will take you to the ORCID site and allow you to create a new iD or authenticate a pre-existing iD in Editorial Manager.

 A3: Corresponding author ORCID ID has been added.

Q4. Thank you for stating the following financial disclosure: 

 [School level scientific research project of Jiangsu Agri-animal Husbandry Vocational College (grant number: NSF2023CB14) . ]. 

 A4: This study received funding from School level scientific research project of Jiangsu Agri-animal Husbandry Vocational College (grant number: NSF2023CB14) . The funders had no role in study design, data collection and analysis, decision to publish, or preparation of the manuscript.

Q5. We note that you have indicated that there are restrictions to data sharing for this study. PLOS only allows data to be available upon request if there are legal or ethical restrictions on sharing data publicly. For more information on unacceptable data access restrictions, please see http://journals.plos.org/plosone/s/data-availability#loc-unacceptable-data-access-restrictions. 

 A5: Sorry, there is a mistake in our description. All relevant data of this manuscript have been included in the manuscript. The content of the statement has been modified as “All relevant data are within the manuscript.”

Q6. In the online submission form, you indicated that [The datasets presented in this article are not readily available, and requests to access the datasets should be directed to ZJ, jing99sam@163.com.]. 

 A6: Sorry, there is a mistake in our description. All relevant data of this manuscript have been included in the manuscript. The content of the statement has been modified as “All relevant data are within the manuscript.”

Q7. Please remove your figures from within your manuscript file, leaving only the individual TIFF/EPS image files, uploaded separately. These will be automatically included in the reviewers’ PDF.

A7: Figures have been removed from the manuscript as requested.

Q8. Please review your reference list to ensure that it is complete and correct. If you have cited papers that have been retracted, please include the rationale for doing so in the manuscript text, or remove these references and replace them with relevant current references. Any changes to the reference list should be mentioned in the rebuttal letter that accompanies your revised manuscript. If you need to cite a retracted article, indicate the article’s retracted status in the References list and also include a citation and full reference for the retraction notice.

 A8: The format of the reference has been proofread and the content has not been modified.

Responses to the reviewers' comments:

Reviewer #1:

Q1: Great study and interesting findings! I have a few suggestions for building some sentences differently:

Abstract: Objective of this study is to investigate...

Line 63: Subsequently histopathological diagnosis is conducted to confirm the nature of the tumor.

L 73: thereby informing about...

L 91: Our aim is to explore...

L 100: Developed by Misdorp et al.

L 185: Analysis of postoperative outcomes revealed...

L 237: The treatment of canine breast tumor is prefereably started in the early stage.

L 246: Should be recognized, diagnosed and treated early.

A1: Thank you very much for your appreciation of the content of this manuscript and your suggestions for specific revisions. We have made detailed modifications according to your modification suggestions, as follows:

“Objective of this study is to investigate the epidemiological characteristics, clinical features, and treatment outcomes of canine mammary tumors in Dalian, providing insights into prevention and management strategies.”

“(3) Subsequently histopathological diagnosis is conducted to confirm the nature of the tumor.”

“This study presents a comprehensive analysis of 198 CMT cases from animal hospitals in Dalian between 2019 and 2023, thereby informing about examining the interplay between breed, age, sex, neuter status, diet, and tumor location.”

“Our aim is to explore the epidemiological patterns and risk factors associated with CMTs, thereby informing more targeted diagnostic, therapeutic, and preventive strategies.”

“Tumor malignancy was classified following the system, developed by Misdorp et al, distinguishing between benign and malignant tumors with malignancy levels graded as I, II, and III [9, 10].”

“Analysis of postoperative outcomes revealed a 100% cure rate following benign tumor resection.”

“The treatment of canine breast tumor is prefereably started in the early and early stage.”

“Therefore, prevention is the most important. Once breast tumor occurs, it should be recognized, diagnosed and treated early.”

Reviewer #2

 Q1: I would say that the statistics unfounded

A1: I am very sorry that the content of this manuscript does not meet your requirements. However, this manuscript is the real case data collected by me in the pet hospital, and all the cases have been objectively analyzed, so as to draw the conclusion of this manuscript.

Q2: Surgical procedure was not explicitly described

A2: I am very sorry for not describing the details of the operation clearly. The detailed description of the operation has been added to the "Surgical procedures" section of the manuscript. Specific modifications are as follows:

“The specific surgical procedures are as follows: First, preoperative examination, anesthesia, and operation in strict accordance with surgical operation specifications. Subsequently, the surgical area was disinfected, the skin was cut, the subcutaneous tissue was removed, and the uterus and ovary were removed first. Then the breast and the skin covering the breast area are removed together according to the above principles of tumor removal. During the operation, the heart rate, blood pressure and respiration were closely monitored. Finally, hemostasis is sufficient to suture the skin and subcutaneous tissue.”

Q3: Qualifications and expertise of the surgeons were not mentioned

A3: Thank you for your comments.The detailed description of the operation has been added to the "Surgical procedures" section of the manuscript. Specific modifications are as follows:

“The operation was performed by Dr. Jing Zheng, who obtained the Chinese Professional Veterinary Doctor qualification Certificate (Certificate number: A012015210149) in 2015. He has been engaged in surgical operations such as tumor resection for 12 years and has rich clinical experience.”

 Q4:Pictures of surgical procedures are not shown to see glitches in the procedure

A4: We are very sorry that only tumor-related pictures described in FIg. 1 were saved in this study, not surgical process pictures. In the future operation process, I will strengthen the preservation of relevant pictures.

---

## [Editor Report · Decision Letter 1]

8 Nov 2024

Epidemiological investigation and surgical treatment of canine mammary tumors in Dalian, China, from 2019 to 2023

PONE-D-24-31787R1

Dear Dr. Jing,

We’re pleased to inform you that your manuscript has been judged scientifically suitable for publication and will be formally accepted for publication once it meets all outstanding technical requirements.

Kind regards,

Sudheesh Sreerangam Nair, B.V.Sc &A.H., M.V.Sc., Ph.D

Academic Editor

PLOS ONE
---

## [Editor Report · Acceptance letter]

13 Nov 2024

PONE-D-24-31787R1 

PLOS ONE

Dear Dr. Jing, 

I'm pleased to inform you that your manuscript has been deemed suitable for publication in PLOS ONE. Congratulations! Your manuscript is now being handed over to our production team.

Kind regards, 

on behalf of

Dr. Sudheesh Sreerangam Nair 

Academic Editor

PLOS ONE